# Older Women’s Experiences of a Community-Led Walking Programme Using Activity Trackers

**DOI:** 10.3390/ijerph18189818

**Published:** 2021-09-17

**Authors:** Jessica O’Brien, Amy Mason, Marica Cassarino, Jason Chan, Annalisa Setti

**Affiliations:** 1School of Applied Psychology, University College Cork, T23 XA50 Cork, Ireland; mcassarino@ucc.ie (M.C.); jason.chan@ucc.ie (J.C.); a.setti@ucc.ie (A.S.); 2Department of Physical Education and Sport Science, University of Limerick, V94 TPX Limerick, Ireland; 17224802@studentmail.ul.ie

**Keywords:** physical activity, technology, user perspectives, older adults

## Abstract

Promoting physical activity amongst older adults represents a major public health goal and community-led exercise programmes present benefits in promoting active lifestyles. Commercial activity trackers potentially encourage positive behaviour change with respect to physical exercise. This qualitative study investigated the experiences and attitudes of older adults following a 6-week community-led walking programme utilising activity trackers. Eleven community-dwelling older women aged 60+ completed individual phone interviews following their involvement in the programme. The programme, codesigned with a group of senior citizens, equipped participants with wrist-worn activity trackers and included biweekly check-in sessions with a researcher to monitor progress and support motivation. Interviews explored participants’ experiences of the programme and of using activity trackers for the purpose of becoming more active. A thematic analysis produced three main themes: ‘programme as a source of motivation’, ‘user experiences with the technology’ and ‘views on social dimension of the programme’. Overall, participants highlighted the self-monitoring function of activity trackers as most beneficial for their exercise levels. This study provides insights into the personal and social factors perceived by older adults in relation to being part of a community-led programme using activity trackers. It highlights the role of the programme and trackers in maintaining motivation to stay active.

## 1. Introduction

Keeping physically active is crucial for healthy ageing [1], with physical inactivity in older adults associated with a plethora of negative health outcomes, including heightened risk of cancer, cardiovascular disease, cognitive decline, depression, falls and fractures [2]. However, staying active in old age can prove challenging. In Ireland, one in three older adults are not meeting the minimum recommended levels of physical activity (i.e., 30 min per day) [3]. Motivating physical activity uptake amongst our older citizens presents a significant public health challenge. Community-led programmes, such as walking groups and exercise in the park, are becoming increasingly relevant in promoting an active lifestyle in older age, with national governments funding such initiatives [4]. These programmes can be run by community health professionals or can emerge as grassroots programmes from organisations of older adults.

Encouraging individual behaviour change is the objective of interventions and programmes which aim to increase physical activity levels. Yet, physical exercise interventions enjoy limited success and low compliance rates in older populations [5]. Wearable technology presents a novel opportunity to promote behaviour change amongst older adults, as self-monitoring tools that operate by bringing one’s attention to one’s own behaviour. The use of wearable activity trackers (e.g., smartwatches and activity trackers) potentially offers a simple vehicle to support behaviour change at an individual level, including for older adults [6,7].

For example, a study with older female participants used Fitbit^®^ devices and support sessions during a 16-week intervention to promote higher physical activity levels [8]. The study equipped participants with activity trackers and an accompanying app to track step patterns and participants were asked to aim for the recommended 150 min of exercise per week. Whether self-monitoring alone was sufficient to motivate greater physical activity was of interest. Increased activity levels post-intervention were found relative to a control group. Yet questions have been raised over older adults’ acceptance of, and adherence to, using activity trackers [9,10], as well as their efficacy, with some studies finding no physical benefits of tracker use in some groups of older adults (e.g., assisted-living adults) [11]. This mixed evidence reflects conflicting findings related to the role of self-regulatory techniques in changing physical activity behaviours (see [12]). French et al.’s [13] systematic review of behaviour change techniques (BCT) for physical activity found that self-monitoring techniques reduced physical activity levels and self-efficacy in older adults, a finding conflicting with a recent review which found self-monitoring approaches effective at reducing sedentary behaviours in adults, including older adults [14]. French et al.’s [13] work indicates the following BCTs as facilitating change in older adults’ physical exercise behaviours; identifying barriers to one’s physical exercise engagement, gaining rewards for exercising and observing others engaged in similar physical activity. Interestingly, French et al. suggest that monitoring one’s own physical activity levels could be too cognitively demanding and/or demotivating for older adults [13]. The mixed outcomes from these reviews illustrate the importance of understanding the use of trackers as enablers of physical activity. Recent studies have explored older adults’ views on using activity trackers for the purpose of increasing physical activity levels.

Ehn et al. [15] investigated older adults’ experiences of using commercial activity trackers over a 9-day period. Participants were offered support in setting up the devices, after which they conducted the intervention independently. They reported participants found self-awareness of their own physical activity as a major positive feature of using activity trackers. Participants in this study reported receiving praise from the device (i.e., getting notifications when a target step count was reached), the usability of the tracker and goal-setting as important factors for motivating and enhancing their experiences [15]. Importantly, participants found the wrist-worn trackers easy to use, whereas they struggled with managing the accompanying tablet app. This speaks to the importance of ensuring the usability of technology incorporated into interventions of this nature for an older population. Qualitative research into older people’s experiences of walking programmes using activity trackers is limited with only a handful of studies to date, and most focus on user acceptance and attitudes towards device design and features [9,16]. A recent review noted the paucity of qualitative research into older adults’ experiences of using activity monitors for their health [17].

In considering activity trackers for promoting physical activity in older adults, the argument has been made that adopting a socio-ecological approach to exercise promotion may be preferrable [18]; environmental factors such as community supports tailored for lifetime stages could represent an important enabler for exercise uptake. This speaks to findings from several studies where social and environmental supports are noted as important motivational factors in older adults’ engagement with regular exercise. For example, Zubala et al. found older individuals’ motivations to exercise increased when the activity was led by their peers [19]. Existing literature on activity trackers in older adults focused on physician-led or expert-led programmes [9,20]. Although this approach is very fruitful to understand which approaches work and which do not, arguably, a community-led approach to the question of activity tracker utilisation within a physical exercise programme for older adults adds an important element: that it is driven by the users themselves. Community-based participatory research (CBPR) offers a way for older adults to contribute as stakeholders to the research process and co-design health initiatives to maximise community uptake and adherence with the ultimate goal of lasting social change [21]. Involving grassroots community groups with older adult members provides a structured means to both include a level of peer support around exercise, an important motivational factor, and secondly, allows older adults to guide decisions around programme format and delivery. Furthermore, as many community-led health initiatives are long-term projects, CBPR offers a valuable way to integrate stakeholders’ feedback and input into future iterations of the initiatives, ensuring such initiatives evolve over time. However, few studies to date explore older people’s experiences of using activity trackers for physical activity as part of community-led or peer-led programmes [22].

Research is needed where behaviour change interventions are designed and delivered in collaboration with established community groups, which would ensure the continuation of interventions beyond the end of the research timeframe. In order to ensure this iterative process, the present study aimed to gather feedback on older adults’ experiences of using activity trackers as part of the community-led physical activity programme which they co-designed and carried out in order to inform future iterations of the programme.

## 2. Materials and Methods

### 2.1. Participants

Fifteen participants who recently took part in the walking programme were contacted. Four individuals were unavailable for interview, resulting in a final sample size of 11. All participants were community-dwelling Irish women, aged between 60 and 80 years, capable of moving independently. All participants were screened for physical fitness and the presence of cardiovascular risk factors prior to engaging in the walking programme, with all participants being relatively active (i.e., meeting the recommended minimum physical activity levels as assessed by the International Physical Activity Questionnaire Short Form (IPAQ-SF) [23]. Purposive sampling was used as all participants had been involved in the 6-week community-led walking programme “Step Up to Your Health”, details of which are provided in Appendix C. The programme was co-designed with a local senior citizens group and involved equipping participants with activity trackers and encouraging them to monitor their walking levels by recording their daily step count in a programme booklet for 6 weeks. Of the 11 participants interviewed, 8 reported no previous experience with activity trackers, 2 reported occasionally using step counting apps on their smartphones, and 1 participant had her own tracking device. Ethical approval for this study was granted by the School of Applied Psychology Research Ethics Committee at University College Cork (Ph.D. 0710201901). The research assistant made telephone contact with participants of the programme to invite them to participate in a follow-up interview. The purpose of the interview was explained to participants, specifically the aim of gathering insights and feedback on their experience of the community walking programme. Verbal informed consent was established with all participants, and they were made aware they could stop the interview at any point. Participants were informed they could withdraw their data from the study up to two weeks after their interview. Participants completed the 6-week walking programme just prior to the national lockdown (a nationwide lockdown was announced the week immediately following Week 6 of the programme). Interviews with participants post-programme were conducted during the lockdown period.

### 2.2. Procedure

Phone interviews were the chosen method of data collection due to the need for social distancing during the COVID-19 pandemic and the accessibility of telephones for older adults. Interviews were conducted on a landline telephone and interview data was audio-recorded using a mobile phone (Huawei P20 Pro, Huawei, China) recoding app, before being transferred to an encrypted laptop and deleted from the phone. Interview data were transcribed verbatim, and the original recordings deleted. Refer to Appendix A for the interview schedule. Interviews lasted approximately 10–15 min.

### 2.3. Approach to Analysis/Theoretical Framework

The qualitative data gathered from the 11 interviews were analysed using thematic analysis. This method of analysis was chosen in line with Braun & Clarke’s guidelines on when to use thematic analysis [24]. Thematic analysis fulfilled our objective of locating patterns in interviewees’ experiences of using activity trackers as part of a community-led walking programme. See Appendix B for further details on our approach to analysis, including a coding tree diagram. The methodological quality of analysis was ensured by the following two measures. Firstly, two researchers (A.M. & M.C.) coded the data independently and agreed the final codes and themes to be included in the report write-up, the themes and related excerpts were discussed with the other co-authors. Secondly, data saturation was agreed based on the principle of diminishing returns [25], whereby data collection was deemed complete once no new themes were emerging from subsequent interview data. Data collection was also closed for pragmatic reasons as our sample was purposive; only individuals who recently completed the walking programme were suitable for interview.

## 3. Results

Data collection generated 19 lower order themes, which were then grouped together to make three higher order themes around participants’ experiences of being involved in an activity tracker-based walking programme: ‘Programme as a source of motivation’, ‘User experiences with the technology’ and ‘Views on social dimension of the programme’.

### 3.1. Programme as a Source of Motivation

The general consensus amongst interviewees was that the community programme did encourage greater levels of walking. Nine out of the eleven interviewees agreed that their involvement in the programme resulted in an increase in the amount of walking they completed in a week. A common thread across the interviews was participants’ describing their behaviour change to highlight how the programme was an effective motivator for them. Being motivated by their involvement in the programme was highlighted by a number of interviewees as playing a major part in their increased walking, with one woman expressing that:

“I was surprised at, you know I would have thought that I was completing ten thousa-in excess of ten thousand steps every day but it just shows you that it, I wasn’t, so it, it’s a good motivator in that way […] and it would push you on.” (P1)

This sense that having the tracker would motivate you to persevere when it comes to keeping active was echoed across many of the interviews. Participants noted that they felt they had changed their behaviour as a result of wearing the tracking device, reasoning that this was either due to having a quantifiable indicator of their daily walking levels through the tracker’s step count or by being motivated by the fact their steps were being monitored by the tracker:

“Yeah just observing my own activity and ehm, becoming more self-aware of how active or inactive I am…I found myself pushing myself a little bit […] so it kind of made me more aware of building up my own stamina.” (P11)

The simple fact of being more aware of personal activity levels had a positive impact on many of the participants. Some referenced that the heightened awareness brought about by using an activity tracker led to lifestyle changes (e.g., walking instead of driving, taking the stairs, parking further away from destination). One lady captured this when she highlighted a small change she had made in her day-to-day life in order to clock up more steps:

“I was watching it and doing my steps and it did encourage me to walk further, park further away in the car park and you know it encouraged me to build up the steps and […] I’d do a bit of walking, bit of exercise to get to a certain figure and so it was very encouraging from that point of view.” (P4)

Furthermore, some participants felt this behaviour change brought about by using the activity tracker was the most important take-away they would learn from their participation in the programme. An insight which points to the long-term benefits of being involved in such a walking programme is that lifestyle changes continued beyond the end of the group.

“In town, I always used to, I stopped using the escalator and I use the stairs. And I still use the stairs now in the shop, I don’t use the escalator anymore so, I thought if I only got just that one thing from it, it was probably worth doing.” (P4)

While the walking programme did not set a target number of steps for individuals to reach, most participants reported being motivated by seeing their step count statistics and wanting to ensure that they reached a certain daily figure which they deemed good for them:

“Now that I’d know if I wasn’t after doing enough of steps, you know I’d try and do something else then to catch up on it, you know.” (P10)

While most participants mentioned the role of step counting in motivating their activity levels, several participants made reference to what the step count represented. High step counts were equated with good health, with one participant mentioning her motivation stemmed from keeping out of hospital by being healthy through keeping her step count consistent:

“When I hadn’t that many steps done, I looked at it as, as something that made me kind of, encouraged me to do something which was for my good and for the good of everybody else as well like, you know? I mean if, if I’m healthy it keeps me out of hospital, leaves more, more space for other people you know that kind of thing, you know?” (P6)

Furthermore, many of the participants reported the tracking device to be of particular use during the COVID-19 lockdown, during which adults aged 65+ were encouraged to remain indoors as much as possible to protect themselves against the COVID-19 virus. Participants found the tracker a useful device to use after the programme ended, when they were trying to stay active without leaving the house:

“I find, I suppose as a matter of interest in the current lockdown, ehm and I’m in the vulnerable section ‘cause I’ve had [an illness] in the last 5 years. So, ahm, I find it’s very useful to keep motivated to do a bit of ehm you know, jogging on the spot or whatever because I’m indoors now for two weeks. So its particularly useful, yeah.” (P8).

While several participants identified the tracker as a source of motivation for walking, some participants felt that neither the programme nor the use of an activity tracker changed their motivation for walking, due to their already fixed physical exercise habits:

“Do you know what? It didn’t make a difference, really, it didn’t. I didn’t miss it when I finished with it, do you know that kind of way?” (P3)

“I suppose I’m not sure I got really into it. Ahm, because I was fairly active anyway. So, it didn’t seem anything different.” (P5)

Some participants commented on not needing external motivation to get out and go for walks, speaking to an inner motivation for experiencing nature and the outdoors:

“I think you don’t need these things to eh, motivate you to get out to the beautiful country side, go for a walk with your dogs, or whoever and I’m all in favour of eh, you know, getting motivation to keep fit and all of that, but I am [70+] and I think that I, I change my way say maybe for a week or two and I say this is great, this is wonderful and then I say, throw it all up in the air.” (P7)

### 3.2. User Experience of the Technology

A common thread across interviews was participants’ experiences as users of the activity tracker being central to the walking programme. The vast majority of participants had not previously experienced using wrist-worn tracking devices. During the programme, participants were briefed on the use of the tracker and had the opportunity to resolve any technical or usability issues by contacting the research team or attending fortnightly check-in sessions. Most participants indicated that the activity tracker device was easy to use and many reported no difficulties with operating it:

“No, I had no difficulties with it, […] I’m not a techy kind of a person, you know? But I, I was able to set the time on it when I went walking and I was able to look and see what my heart rate was and […] How many steps I had done and stuff, you know I didn’t have a problem with it.” (P11)

A number of participants also reported that the device was “comfortable to wear” (P4). Aside from the step counting function of the tracker, other positive features of the device were commented on by group members. In particular, the sleep pattern analysis was highlighted as an interesting feature by a number of participants:

“Well the benefit was I used to think I was a very bad sleeper, but now I realise actually I get quite a lot.” (P2)

In addition, the alert function of the tracker to encourage participants to walk at least once per hour was utilised by some participants and seen as a worthwhile aspect of the device:

“I also put an alert on my [tracking device] to, you know it beeps three times if I’ve been sitting down for an hour. So, that makes me get up and do a few jumps or whatever you know? […] and I found that was a good part of it as well.” (P8)

Most participants required help with the initial setting up of the devices, but once initial setup was complete, they generally found the devices easy to use. Many of the women mentioned their unfamiliarity with such technology, describing themselves as 

“not techy” yet still finding the device useful and reporting a positive user experience. P1 sums this sentiment up: “[I’m] not big into gadgets and that, but I think that […] it is a worthwhile bit of equipment.” (P1)

In contrast to some participants’ views of the usability of the device, technical issues with the tracker made the experience less seamless for others. Technical issues reported by interviewees included: the device not charging, the device not being synced to the participant’s phone and the device deleting the day’s data at a certain time each day:

“I didn’t really understand how to use it [the device] properly, I’d be able to count the steps and then the steps would […] be gone, and there was days I, I didn’t know how many steps I had, I was only judging it using an average.” (P4)

A number of participants felt “a bit restricted” (P1) by the particular model of activity tracker being used and some felt that it did not capture the full extent of their completed exercise:

“What I did find about the exercise was it only records steps, you know. I was, we say now in aqua aerobics, I would be doing a lot of movement in that but you’re not recording anything in that kind of exercise, you know? It didn’t really sort of ehm, describe how, how much exercise I was doing.” (P4)

For those participants who shared a positive engagement with the activity tracker, some bought their own tracker after the programme or expressed an intention to do so. Seven of the eleven participants reported that they intended to continue using an activity tracker beyond the end of the programme. Interestingly, many participants described feeling curious about their activity levels during the COVID-19 lockdown, during which time the programme had ended and most individuals no longer had an activity tracker:

“No, I didn’t [buy an activity tracker] and actually I was tempted, I was saying to myself do you know now with this lockdown and everything […] you can only go out once a day and the rest of it, I would have gone on longer walks, but now I can just do […] the two kilometres and what have you. So, I’m kind of more restricted and I would have found it more interesting to see how much movement I was doing, you know?” (P3)

### 3.3. Views on Social Dimension of Programme

Speaking to the importance placed on maintaining social connections in later life [1], commentary on the social dimension of the ‘Step Up to Your Health’ programme was prevalent across the interviews. The social aspect of the programme involved meeting as a group fortnightly in an active retired meeting space to check-in with participants regarding any technical issues and to provide encouragement to continue regular walking habits. In terms of participants’ reflections on the social elements of the programme, two narratives emerged regarding the value of social connections within the programme. It appears a balance needed to be established regarding the pros and cons of including social aspects. Several participants expressed an interest in, and even an expectation of, having a greater social component to the programme. One lady provided an insight into this perspective, indicating that, for her, creating social connections during the programme would be crucial to her enjoyment of and engagement with the walking programme:

“I thought that we’d be walking as a group […], I was hoping like that, you, I’d be motivated because I’d have to start at a particular time and meet people and do it that way, you know? […] the [tracking device] itself didn’t do anything for me” (P5)

This participant highlighted some individuals’ expectations of and wishes for a greater social element to the programme, considering that a walking group was more motivating for her and that she perceived little motivation from using the device alone. Several participants echoed this call for a greater provision of interaction with peers during the programme, highlighting the limited opportunities to meet as a group to discuss and compare activity levels (i.e., steps):

“No, I didn’t [compare steps] because we seemed to go in kind of individually, do you know what I mean?” (P3)

“So in the normal situation, I think, it, it would be quite interesting to be with a group because you’d be discussing your activity and you’d get ideas and tips from people, you know? And so, I think it, it would be very interesting to use as part of a group, but it was certainly very interesting to use it on an individual basis as well.” (P11)

In contrast to some participants’ desire for a strong social component to the programme, other participants revealed a level of discomfort with the social aspect of the group. A number of participants referred to being averse to any comparison of steps and shared their experience of feeling intimidated by the activity levels of others in the group:

“When she arrived in one morning and I think she’d already done about five or six thousand steps, oh God, put me to shame, so I wouldn’t be able to compete with them like.” (P2)

This participant offered a different perspective on the social element of a group programme, speaking to a feeling of embarrassment when comparing steps to peers, even if the comparison was self-initiated and not part of the programme. Given a number of participants’ wishes for a stronger social element to the programme (specifically the idea of a walking group), P2’s point on the negative aspect of comparison of steps highlighted the need to find a balance when considering the social dimension of such programmes. Increased social connections or social support during the programme also posed more opportunities for the comparison of steps and may have negatively influenced motivation levels for some individuals. Interestingly, while only one participant voiced discomfort at comparing step counts, others commented on a similar lack of interest in comparing step counts amongst the group, preferring using the device as a personal challenge:

“It would be something I’d do for myself, I wouldn’t be competitive or anything like that.” (P3)

## 4. Discussion

This qualitative study gathered accounts of a group of older women’s experiences of being part of a community-led walking programme utilising activity trackers. The three overarching themes generated from the interviews revealed participants’ experiences centred around the programme as a source of motivation for behavioural change, how familiarity with technology shaped their enjoyment of the programme, and the complexity of the social dynamics of a community-led walking programme.

In general, participants reported the activity tracker as a source of motivation to exercise, with seven out of the eleven participants considering purchasing or having purchased their own activity tracker after the programme. Our findings corroborate previous research indicating older adults’ acceptance of tracker devices being dependent on the degree to which the device can be useful for them [15]. The four participants who did not intend to continue using activity trackers in their daily lives beyond the programme, did not perceive the device to be of use to them, either due to frustration with technical issues or finding the device was not personally motivating for them. The Unified Theory of Acceptance and Use of Technology [26] predicted such patterns of behaviour with the perceived usefulness of technology being equated with the continued use of the technology. As an exemplar of this theory, one participant in our study described not “taking to” the device because she found herself reverting back to what she has always done and that the activity tracker failed to motivate her in a new way because simply counting steps did not motivate her and she did not utilise the other functions available. In short, the device did not meet her needs and so she discontinued using it [27]. Other reasons participants would not continue using a tracker were related to the usability of the device or the specific model used in this walking programme, specifically, not being able to consistently operate the device or feeling frustrated that the tracker could only capture steps and not other physical movements completed by participants (e.g., aqua aerobics, as specified by one participant). By contrast, those who intended to continue using the tracker post-programme were participants with positive experiences of using the activity tracker and reported feeling motivated. Future research and walking programmes aimed at improving the acceptability of trackers for older people should consider the usability of devices for activities older adults may favour but which are not captured as steps on trackers (i.e., physical exercise with less vigorous movements).

Our findings contribute to the growing qualitative literature on older adults’ experiences of using wearable tracker technology for the promotion of physical activity with the novel element of the utilisation within a community-led programme. Considering our group of participants as a whole, devices were well-accepted, with most reporting a predominantly positive experience. This is in line with qualitative insights from Tully et al.’s [22] peer-led walking programme, where pedometers were used and well-accepted by older participants. The personal experiences of most participants echoed past literature documenting that simply tracking one’s physical activity levels was sufficient to reduce sedentary behaviours, which many of our participants experienced during the walking programme [14]. Our qualitative work weighs in on the conflict in the literature with regard to the effectiveness of self-monitoring techniques to encourage greater physical exercise amongst older people [13]. The self-monitoring purpose of activity trackers was specifically highlighted as beneficial by participants. Most participants experienced that having a ‘realtime’ record of their activity levels via the tracker as intrinsically motivated their walking levels. Quantifying their walking behaviour into steps made them self-aware of their activity levels which, in turn, encouraged them to increase their step metrics. Importantly, those individuals who experienced little benefit from being involved in the programme reported being already self-aware of their personal activity levels and the benefits of spending time moving outdoors. This speaks to the primary role of the activity tracker as a self-monitoring device and the benefit of this behaviour change technique in some older adults, many of whom have little experience with such technology or other forms of monitoring one’s activity levels. The effectiveness of self-monitoring to bring about behaviour change in older adults is likely dependent on self-awareness or the previous engagement of the individual with self-tracking their exercise, as well as whether they are more intrinsically or extrinsically motivated. This may account for the mixed results in reviews on self-monitoring as a means to instigate behaviour change in older adults [13], despite considerable evidence that older adults find activity trackers enjoyable and motivating [15,16].

In term of the programme itself, and the usefulness of the trackers within the programme, several participants revealed their expectations for a significant social element to the programme and a number of interviewees highlighted a desire for a greater social aspect to the group. Specifically, group members mentioned that providing greater opportunities for social interaction and the formation of social walking groups would enhance the value of the programme for them. This sentiment within the group reflects previous research indicating that social interaction can be a strong motivator for exercising [28]. For example, in Tully and colleagues’ pilot RCT, intervention participants reported that walking with their peer mentor was both an enjoyable and an important motivator for them to exercise [22]. The biweekly check-in sessions throughout the course of the 6-week programme were intended to fulfil the social element of the study; however, these were semi-structured and predominately focused on participants sharing their experience of monitoring their steps and using the tracker device. To address this element of the programme in the future, small walking groups within the larger group could be formed so that its members could agree on a specific time and location for walking, where they could discuss the technology and share ideas. This would add further structure to the programme and allow for a relaxed discussion as opposed to addressing individual technical issues in a group setting only. Interestingly a juxtaposition emerged, whereby some women wished for greater social engagement as part of the programme. Alternatively, the experiences and views of others in the group indicated that social comparisons in this aspect can lead to discomfort and embarrassment. These conflicting viewpoints between participants pointed to a trade-off at play in group programmes such as this one; the value of a strong social dimension and the possible negative consequences of this, including social comparisons and feeling self-conscious about one’s activity levels. A balance needs to be established to ensure participants have enough opportunities for social engagement to increase the value of the programme and interest from this age group, but also to ensure this is not to the detriment of participation and enjoyment, as social comparisons may impact satisfaction levels and positive wellbeing outcomes.

A unique aspect of this study is that we gained an insight into how wearable activity trackers can be useful to older adults when in lockdown and with limited movement advised outside the home (at the time of interview, over 70s were advised to avoid contact with others and travel/exercise was restricted to within 5 km of one’s home). Many participants expressed their gratitude for having the device during the national lockdown and others who had since returned their device conveyed a curiosity and desire towards the benefits of using it during those times. Wearing the device can aid motivation and accountability to exercise, despite the change in routine due to the COVID-19 outbreak. The importance of older adults continuing physical activity during lockdown cannot be overstated, not only for physical health but also to combat the psychological impact of being isolated [29]. This study also provided an insight into how older adults chose to exercise when leaving home was not an option. Most participants reported trying to accumulate steps by exercising around the house, gardening, cleaning, walking up and down stairs, or jogging on the spot, corroborating reports that low-resistance exercise is preferred amongst this age group [28].

### 4.1. Strengths and Limitations

This study aimed to enrich a growing area of research by considering older people’s experiences of using technology as part of a community-led programme for physical activity. The study saw participants as co-creators of the walking programme. Considering the purposive sample, the study had a satisfactory response rate.

The study is not without limitations. Noteworthy limitations pertain to the COVID-19 outbreak in March 2020. Due to the necessity of social distancing when the post-programme interviews were conducted, these interviews had to be completed via telephone. This posed difficulties for data collection; the inability to use non-verbal cues, disruptive phonelines leading to an inability to contact, and thus include four participants. Phone interviews instead of in-person interviews may also have impacted the willingness of participants to elaborate further on their experiences. Secondly, during the COVID-19 pandemic, in Ireland, individuals over the age of 70, or those with underlying health conditions were required to self-isolate at home from mid-March, meaning substantial changes to their daily routines, including limited options for physical exercise (e.g., restricted to areas close to home). However, this was not considered a major setback to the research as the experience of using trackers and of the programme could still be evaluated remotely.

Participants were exclusively female and therefore findings may differ for a male population. It is possible that males could have differing opinions and experiences of the monitoring device. However, as Tully et al. also found in their CBPR walking study [30], a disproportionate amount of females over males engage in community walking programmes and our research reflects this naturally occurring gender difference in community exercise participation. In addition, some of our participants had previous experiences of using activity monitoring devices; two participants previously used phone apps for step counting and one used her own wearable device. It is likely that these participants were more familiar with activity monitors, and thus were less likely to run into user difficulties, increasing the likelihood of a positive experience and the long-term adaptation to the device compared to those unfamiliar with activity trackers. However, as expected they were not likely to experience any change in behaviour or motivation as this would have occurred with their previous device.

In terms of the interviews themselves, only post-programme interviews were conducted; therefore, no formal gathering of opinions and/or perceptions of the technology, before the programme were collected. It is recognised that pre-programme perspectives of the technology would have been useful for comparison purposes. The capabilities of activity monitoring devices should be customisable to the activities that older adults are most likely to partake in and should include the option to track low-impact activities like gardening and cleaning. This was a short-term study of 6 weeks’ duration; long-term studies are scarce and would provide much needed insight into how the user experience of wearable trackers changed over time and to explore whether such devices could be successfully adopted by older adults in the long term.

### 4.2. Future Directions

Continued research in this field of investigation is paramount to provide further insights aimed at supporting older adults in staying active and healthy. Based on the accounts from the participants in our study, additional research may benefit from focusing on two aspects of the programme design to maximise older people’s engagement and enjoyment of community-led tracker programmes: ensuring a strong social component to the programme and ensuring adequate ongoing support for arising technical issues with the tracker. Incorporating a strong social element may pose a challenge, given insights from our study that the group element of the programme may jeopardise motivational levels, with isolated accounts of negative self-perception of one’s own exercise levels. Finding ways to ensure social connections during the programme design without inducing comparisons between participating individual’s exercise goals should be the focus of future studies adopting a community programme of this nature.

## 5. Conclusions

In conclusion, we present a novel investigation into older adults’ personal experiences of engaging in a community-led walking programme centred around using activity trackers. The findings from this study highlight the positive experience for older adults using activity monitoring devices in a community setting, specifically, older adults’ views of the tracker device as a source of personal motivation, allowing them to quantify their daily activities and engage in personal goal setting. We highlight the use of activity trackers in a community group setting as a viable and novel programme approach for physical activity promotion efforts.

## Data Availability

Due to the small sample of participants and qualitative nature of the data, the raw data (interview transcripts) will not be made publicly available. Anonymised versions of the transcripts are available upon request from the authors.

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
