# Peer review of "Older Women’s Experiences of a Community-Led Walking Programme Using Activity Trackers"

_ijerph, 2021, doi:10.3390/ijerph18189818_

Round 1

Reviewer 1 Report

I have a few humble opinions.

  1. CBPR is a collaborative research approach in which community members participate. Participants are collaborators and decision makers. Another advantage is that it can be continued without interruption after the program is over. However, the merits of these community programs of CBPR were hardly revealed in the intro part.

2. It is necessary to explain why thematic analysis was selected as a qualitative data analysis method. According to Braun and Clarke, thematic analysis may not meet its rigor because of its flexibility. This is a disadvantage. It can also be a weak analysis.

3. 6 weeks is not enough to confirm the effect of exercise. Although the initial effect of exercise begins to appear at 4 or 6 weeks, it is ideal for at least 8 or 12 weeks. Therefore, it is necessary to describe whether the reason for setting the period of 6 weeks in this study is due to lockdown or the original design. The rationale for setting 6 weeks as a period should be added. 

4. Also, since the interview time is about 10 to 15 minutes, it is questionable whether data saturation is possible. Including the introductory question, the questions should be systematically described to improve the readability of the reader (in Appendix A). Even if the interview time is short, it should be stated whether sufficient data have been collected.

5. In result, brief information about 11 subjects is needed. In particular, it is necessary to check whether the group is a similar group capable of walking. Information is needed on the subject's musculoskeletal disease, chronic disease.

6. In the results, ‘Program as a source of motivation’, behavior change is expressed together with the motivation. It needs to be analyzed separately.

7. The 'User experiences with the technology' part of the result is meaningful to check the essence of how the new experience with this equipment is different from the previous experience. It is also suggested to analyze the strengths and weaknesses, past experiences, and other aspects in more detail.

8. In the discussion, there is a need for further discussion of the frustration experience of 4 participants. They may discover new challenges or solutions through their experiences.

In addition, it is necessary to discuss the CBPR effect of other studies.

Reviewer 2 Report

Thanks a lot for this valuable paper! All parts of the manuscript are generally of acceptable/high quality. As for the Introduction, I suggest to add references in some places and to check some additional references I provide below. Especially the reviews might provide valuable additional insight into the topic of interest. As for the Materials and methods, some additional information about the selection and motivation of the participants is needed. The Results are presented in an adequate way and are in the further well discussed in the Discussion section. Here, I suggest to add some aspects to the limitations. The Conclusions are well based on the discussion of the results.

Please find below my specific comments:

Introduction

l. 33-37: I suggest to support these assumption and potential by references.

l. 103-105: Please check the following references:

Franssen, W.M.A., Franssen, G.H.L.M., Spaas, J. et al. Can consumer wearable activity tracker-based interventions improve physical activity and cardiometabolic health in patients with chronic diseases? A systematic review and meta-analysis of randomised controlled trials. Int J Behav Nutr Phys Act 17, 57 (2020). https://doi.org/10.1186/s12966-020-00955-2

Sullivan, A. N., & Lachman, M. E. (2017). Behavior Change with Fitness Technology in Sedentary Adults: A Review of the Evidence for Increasing Physical Activity. Frontiers in public health, 4, 289. https://doi.org/10.3389/fpubh.2016.00289

Tully, M. A., Cunningham, C., Wright, A., McMullan, I., Doherty, J., Collins, D., Tudor-Locke, C., Morgan, J., Phair, G., Laventure, B., Simpson, E., McDonough, S. M., Gardner, E., Kee, F., Murphy, M. H., Agus, A., Hunter, R. F., Hardeman, W., & Cupples, M. E. (2019). Peer-led walking programme to increase physical activity in inactive 60- to 70-year-olds: Walk with Me pilot RCT. NIHR Journals Library.  

Materials and methods

l. 110-111: Why exactly these 15 participants were contacted? Why were they chosen? Why did these 11 agree to participate? What was their motivation to participate? These are important information missing at the moment. The results have to be discussed e.g. towards the motivation of the participants to join the study. 

l. 120-121: Please provide information about consent and withdrawal options for participants. l. 130-132: Please describe how participants were briefed about the interview procedure, including information about consent/withdrawal.  

Results no specific comments  

Discussion  

4.1 Strengths and Limitations:

Besides the fact that the sample was exclusively female and that some of the participants had prior experience in using activity monitoring devices, it should also be discussed in how far the selection of the participants and the reasons why they decided to take part could have had an impact on the results.

Further on, it should be noted that the sample is very small, with only 11 participants.

Conclusions

No specific comments

Round 2

Reviewer 1 Report

As a result of reviewing the revisions made by the researchers, it is judged to be reasonable. However, additional explanations about how factors that can affect exercise was controlled seem necessary.
